# Experimental animal models and their use in understanding cysticercosis: A systematic review

Muloongo C. Sitali[1]*, Veronika Schmidt[2], Racheal Mwenda[3], Chummy S. Sikasunge[3], Kabemba E. Mwape[4], Martin C. Simuunza[5], Clarissa P. da Costa[6], Andrea S. Winkler[7], Isaac K. Phiri[4]

1 Department of Biomedical Sciences, School of Veterinary Medicine, University of Zambia, Lusaka, Zambia, 2 Centre for Global Health, Institute of Health and Society, University of Oslo, Oslo, Norway, 3 Department of Paraclinical Studies, School of Veterinary Medicine, University of Zambia, Lusaka, Zambia, 4 Department of Clinical Studies, School of Veterinary Medicine, University of Zambia, Lusaka, Zambia, 5 Department of Disease Control, School of Veterinary Medicine, University of Zambia, Lusaka, Zambia, 6 Institute for Medical Microbiology, Immunology and Hygiene, Technical University of Munich, Munich, Germany, 7 Department of Neurology, Centre for Global Health, Klinikum Rechts der Isar, Technical University Munich, Munich, Germany

* sitalimuloongo@yahoo.com

**Data Availability Statement:** All relevant data are within the paper and its Supporting Information files.

## Abstract

### Background

Cysticercosis and Neurocysticercosis (NCC) can be studied using several animal species in experimental models which contributes to the understanding of the human form of the disease. Experimental infections of *Taenia* spp. are vital in explaining the modes of transmission of the parasite and helps the understanding of transmission of the parasite in humans and thus may be useful in designing therapeutic and immune-prophylactic studies to combat the disease. Thus, this systematic review aims to explore the existing experimental animal models to the understanding of cysticercosis in both humans and animals and elucidate the risk factors of cysticercosis and identify the *Taenia* spp. used in these models.

### Methodology

We systematically identified all publications from the Web of Science, Google Scholar, and Pubmed regarding experimental animal models using *Taenia* spp. that cause cysticercosis in both humans and animals. 58 studies were identified for eligibility. Of these, only 48 studies met the inclusion criteria from which data extraction was done and presented descriptively.

### Results

Pigs, cattle, gerbils, mice, rats, voles, monkeys, cats, dogs, and goats were used in which *T. solium*, *T. saginata*, *T. saginata asiatica*, *T. crassiceps* and *T. asiatica* were studied. The routes used to induce disease were; oral, intravenous, subcutaneous, intramuscular, intraperitoneal, intraarterial, intracranial, intraduodenal, and surgical routes using eggs,

**Funding:** The author received no specific funding for this work.

**Competing interests:** The authors have declared that no competing interests exist.

oncospheres, and proglottids. Besides, the establishment of infection using eggs and oncospheres was affected by the route used to induce infection in the experimental animals. The cysticerci recovery rate in all the experimental studies was low and the number of animals used in these experiments varied from 1 to 84. Although not analysed statistically, sex, age, and breed of animals influenced the cysticerci recovery rate. Additionally, the cysticerci recovery rate and antibody-antigen levels were shown to increase with an increase in the dose of oncospheres and eggs inoculated in the animals. Contrasting results were reported in which the cysticerci recovery rate decreased with an increase in the dose of eggs inoculated.

## Conclusion

This review describes the various animal experiments using *Taenia species* that cause cysticercosis highlighting the animals used, age and their breed, the routes of infection used to induce disease and the sample size used, and the cysticerci recovery rate in these animal models.

## 1.0 Introduction

The larval stage of the Cestoda family *Taeniidae* known as the cysticercus causes an infection called cysticercosis [1]. The cysticerci are recovered from various tissues in the infected animals and the cysticerci recovery rate can be defined as the number of cysts recovered from the infected animals depending on the dose of the infecting material used [2]. Cysticercosis is a parasitic zoonosis that has been ranked on top of the Neglected Tropical Diseases (NTDs) list by the World Health Organization (WHO). The institution of appropriate control measures, however, still requires further research [3].

The genus *Taenia* contains many species that infect humans and domestic animals. Of these, *T. solium* and *T. asiatica* eggs can infect pigs if they ingest eggs excreted from human tapeworm carriers [6] whereas cattle serve as intermediate hosts for *T. saginata* [1]. In addition, small ruminants such as mice, rabbits, and other rodents serve as intermediate hosts for the larval stage of *T. crassiceps* which share the definitive hosts (i.e. dogs, foxes, wolves, and felids) with *T. hydatigena* [4, 5].

*Taenia saginata*, *T. solium* and *T. asiatica* share the same definitive host (humans). Eggs or gravid proglottids are passed with faeces of humans and cattle (*T. saginata*) or pigs (*T. solium* and *T. asiatica*) become infected by ingesting vegetation contaminated with eggs or gravid proglottids [6, 7]. The oncospheres hatch in the animal's intestines, invade the intestinal wall and migrate to striated muscles, where they develop into cysticerci (*T. solium* and *T. saginata*) [6, 7]. However, *T. asiatica* larval stage attacks the visceral organs of the pig [8]. Humans become infected by ingesting raw or undercooked infected meat (*T. solium* and *T. saginata*) or infected liver (*T. asiatica*) [6]. About 2–4 months, a cysticercus develops into an adult tapeworm in the human intestine where they attach to the small intestine by their scolex. Accidental ingestion of *T. solium* eggs causes cysticercosis and neurocysticercosis in humans [9]. In contrast, it is not yet postulated whether *T. asiatica* causes hepatic cysticercosis in humans [8].

The life cycle of *T. crassiceps* starts in the intestines of wild carnivores where it reproduces [10]. The infective eggs are released in the faeces of the carnivores which are eaten by rats [11]. The life cycle repeats when rats harbouring the larval stage are eaten by another canine.

Humans are rarely infected by *T. crassiceps*, if infection occurs, it causes ocular larva migrans which may result in blindness especially in immune-compromised individuals [11].

Cysticercosis caused by *Taenia* spp. affects several species including humans, cattle, goats, sheep, pigs, and dogs. However, cysticercosis in humans and pigs caused by *T. solium* is an important socio-economic problem in countries where poverty, poor sanitation, and hygiene prevail which usually favour transmission of the parasite. In humans, the most frequent form of the disease is neurocysticercosis (NCC) [12, 13].

The processes that occur during the infection course in cysticercosis and NCC can be studied in animal models that closely resemble the parasite life cycle [14, 15]. This is useful to the understanding of the pathophysiological processes, identification of specific biomarkers for early stages of development, the immune response, and pathological outcomes. Furthermore, animal models with high rates of viable cyst infections in skeletal muscles, brain, subcutaneous tissues, lungs, eyes, liver, and the heart, thyroid, and pancreas may control for variables such as infection dose [16]. Moreover, experimental models are useful to the comprehension of the host-parasite relationship and thus aid in understanding cysticercosis in both humans and animals in detail [12].

Animal models are useful for investigating the process that occur during the infection course of cysticercosis in various animal species [16]. These models help to further understand the immune response mounted by animals and may aid in the development of vaccines and help in the identification of specific biomarkers for development of disease [16]. Moreover, experimental studies of *Taenia* species may help in the testing of vaccines in order to interrupt the life cycle of *the parasites* by preventing animals from obtaining the larval stage [13].

The purpose of this study was to provide a systematic review of the existing experimental animal models (species or strain of animals used, sample sizes used, and the method of inducing disease) and the risk factors of cysticercosis. These are important because they contribute to the understanding of the human form of the disease. Additionally, the study aimed to investigate the cysticerci recovery rate in the animal models which may help future cysticercosis research experimental designs if desired results are to be obtained.

## 2.0 Methods

In this study, all experimental animal models using *Taenia* spp. that cause cysticercosis and met the inclusion criteria were included. The systematic review was conducted according to the Preferred Reporting Items for Systematic Reviews and Meta-Analyses (PRISMA) guidelines in March 2021 [18, 19]. Information was collected on the methodology of the model (Species of animals used, infection method, infecting material used, sample sizes, sex, age, and duration of the study) model strengths and/or weaknesses, and relevant outcomes of the study. We did not register our systematic review protocol on the database PROSPERO as our data extraction process was already completed by the time we obtained knowledge on how to register the protocol on PROSPERO.

### 2.1 Eligibility criteria

**2.1.1 Inclusion criteria.** Articles that met the following criteria were included: studies concerning *Taenia* spp. causing cysticercosis, animal model studies, experimental studies, studies conducted between 1st January 1980 and 11th October 2021. This followed the PICOS (population, intervention, comparison, outcome, and study design). (1) population- all *Taenia* spp. causing cysticercosis in experimental models; (2) Intervention- all infection methods in experimental models using *Taenia* spp.; (3) Comparison- the animals not infected with either eggs, proglottids or oncospheres; (4) Outcome- infection status of the animals, either

cysticercosis positive or negative; (5) study design- experimental study designs involving *Taenia* spp. causing cysticercosis in animals.

   **2.1.2 Exclusion criteria.**   Publications were excluded if at least one of the following criteria was met: (1) Studies did not concern *Taenia* spp. causing cysticercosis; (2) Studies were not animal models; (3) Studies were not experimental; (4) Studies conducted before 1st January 1980 or after 11th October 2021; (5) Studies with results outside the scope of the study questions (including general reviews on the topic).

## 2.2 Information sources

The information was obtained from online databases; Web of Science, PubMed, and Google Scholar.

## 2.3 Search

The search was conducted in the three databases between 1st and 4th Dec 2020 using one search phrase, another search was conducted on 11th October 2021. The search phrase read as follows; (Experimental infection OR experimental model OR animal models OR animal experiments) AND (Cysticercosis* OR neurocysticercosis*) AND (*Taenia solium* OR *Taenia saginata* OR *Taenia ovis* OR *Taenia hydatigena* OR *Taenia crassiceps* OR *Taenia asiatica*) AND (Infection using eggs OR infection using oncospheres OR infection using proglottids).

## 2.4 Study selection

The PRISMA guidelines were used to select studies. Duplicates were removed from the total publications searched. The remaining publications were screened on title and abstract and those that did not meet the inclusion criteria were excluded. Full texts were then read and those that met the inclusion criteria were included for data extraction.

## 2.5 Data collection process

Data extraction included methodology of the model (Species of the animals used, infection method, sample sizes, infecting material, and age of the animals used) model strengths and/or weaknesses, and relevant outcomes of the study. Data extraction was done by one reviewer (MCS) while the other reviewer verified the extraction (RM). Other reviewers were consulted where there was disagreement. Data was entered in the excel spreadsheet.

## 2.6 Data items

The following data was extracted; Species of animals used, *Taenia* spp. used, age of the animals used, number of the animals used, breed or strain of the animals used, sex, author and year of publication, duration of the study, method of infection (oral, surgical or any other method employed in the study), infecting material used (eggs, proglottids and oncospheres), and the cysticerci recovery rate.

## 3.0 Results

### 3.1 Study selection

The computerized search yielded 999 articles, of which 857 were retained after removing duplicates. Based on title and abstracts, 781 titles were removed for being non-experimental studies. Data on 48 articles were included in the extraction (Fig 1 and Tables 1–8).

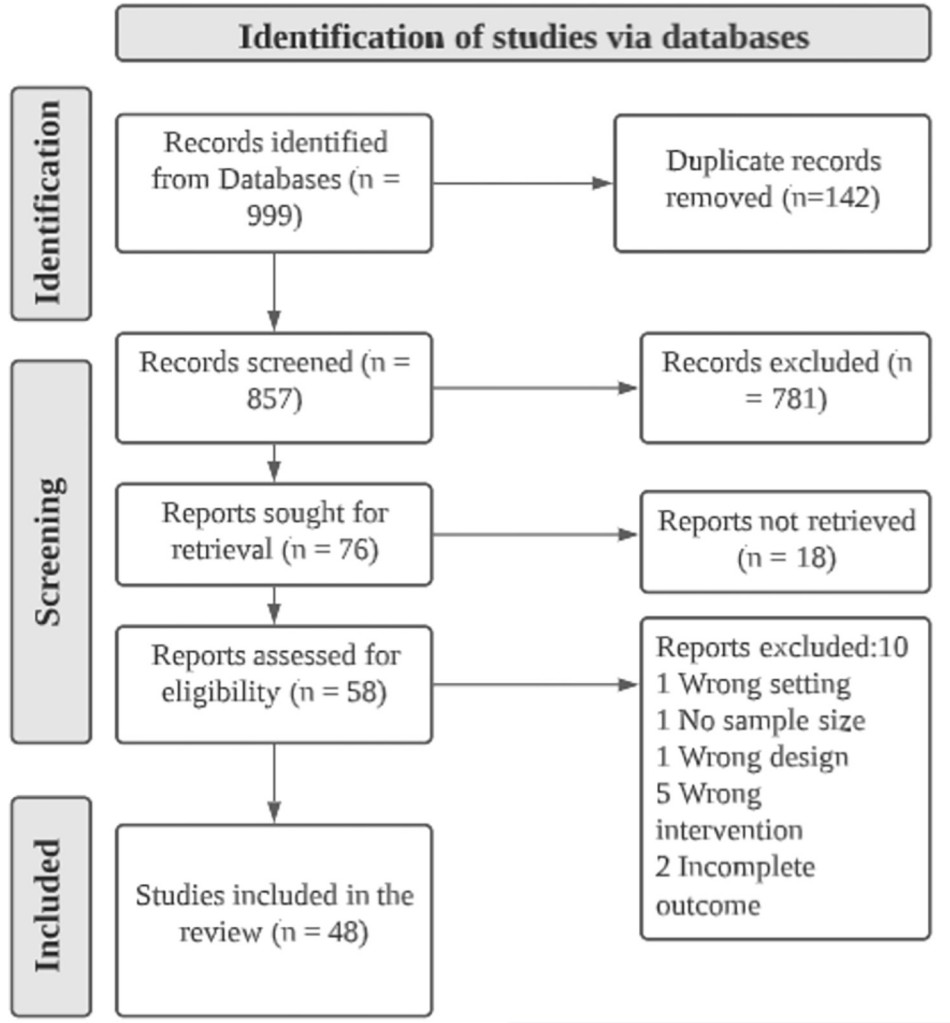

**Fig 1. PRISMA flow diagram.**

## 3.2 Risk of bias assessment

The Systematic Review Centre for Laboratory animal Experimentation (SYRCLES's) risk of bias tool for animal studies [61] was used to assess the risk of bias for all studies included in the review after the full-text screening (Fig 2). Two independent reviewers assessed the risk of bias in the included studies. Disagreements between reviewers were resolved through consensus. The assessment of the risk of bias showed that there was no blinding of investigators involved in enrolling participants (animals) and they could foresee the assignment of interventions and thus introduce selection bias. Moreover, there was no randomization in the selection of subjects.

## 3.3 Quality assessment of the studies included

Quality assessment of included studies was performed independently by two reviewers, blinded to the name of the authors. The quality of included studies was assessed by the two reviewers according to the Animal Research: Reporting In Vivo Experiments (ARRIVE) guidelines Checklist [62].

**Table 1. Description of the animal models used in the included studies for *T. solium* and *T. s. asiatica*.**

| Author | Animals used | Sample size | Taenia spp. used | Cysticerci recovery rate/density |
|---|---|---|---|---|
| Carmen-Orozco *et al.*, (2021) [20] | rats | 19 | *T. solium* | Not Reported |
| Palma *et al.*, (2019) [21] | rats | 36 | *T. solium* | Not Reported |
| Mejia *et al.*, (2019) [17] | rats | 100 | *T. solium* | 7–32 cysts |
| Alroy *et al.*, (2018) [16] | pigs | 12 | *T. solium* | 0.64–1.14% |
| Gomez-Puerta *et al.*, (2018) [22] | pigs | 20 | *T. solium* | 2–40 cysts per pig |
| Fleury *et al.*, (2015) [23] | pigs | 24 | *T. solium* | 3.6–5.4% |
| Verastegui, *et al.*, (2015) [24] | rats | 84 | *T. solium* | 1–4 cysts per rat |
| Borkataki *et al.*, (2013) [25] | pigs | 5 | *T. solium* | Not Reported |
| Da Silva *et al.*, (2012) [26] | pigs | 7 | *T. solium* | 3.78–81.93% |
| Peng *et al.*, (2009) [27] | mice | 15 | *T. s. asiatica* | Not Reported |
| Deckers *et al.*, (2008) [28] | pigs | 18 | *T. solium* | 1.5–98.5% |
| Maravilla *et al.*, (2008) [29] | pigs | 4 | *T. solium* | 0.2–4.2% |
| Garrido *et al.*, (2007) [30] | pigs | 12 | *T. solium* | Not Reported |
| Linghu *et al.*, (2007) [31] | pigs | 12 | *T. s. asiatica* | Not Reported |
| Fan *et al.*, (2006) [32] | pigs, cattle, goats, monkeys | 83 pigs, 10 calves, 17 goats, 4 monkeys | *T. s asiatica* | 0.005–22% pigs, 0.03–6% calves, 0.01–0.02% goats, 0.01% monkeys |
| Soares *et al.*, (2006) [33] | pigs | 7 | *T. solium* | 0.42–28.57% |
| Chang *et al.*, (2005) [34] | mice, hamsters, gerbils | 18 mice, 55 hamsters, 81 gerbils | *T. s. asiatica* | 0.1–3% mice |
| Nguekam *et al.*, (2003) [35] | pigs | 14 | *T. solium* | 0.03–3.2% |
| Liu *et al.*, (2002) [36] | mice | 80 | *T. solium* | 1–2 cysts per mouse |
| Santamaria *et al.*,(2002) [37] | pigs | 30 | *T. solium* | 0–2.5% |
| Verastegui *et al.*, (2002) [38] | pigs | 4 | *T. solium* | 0–138 cysts per animal |
| Verastegui *et al.*, (2000) [14] | pigs | 18 | *T. solium* | 0–69 cysts per animal |
| Wang *et al.*, (2000) [39] | mice | 20 | *T. s. asiatica* | 0.3–4.6% |
| De Aluja *et al.*, (1999) [40] | pigs | 13 | *T. solium* | Not Reported |
| Wang *et al.*, (1999) [2] | mice | 10 | *T. s. asiatica, T. solium* | 2.4–3% |
| Ito *et al.*, (1997a) [41] | mice | 18 | *T. s asiatica, T. solium* | Not Reported |
| Chung and Fan, (1996) [42] | pig, monkey | 12, pigs, 1 monkey | *T. s. asiatica* | 1.9–22.9% pigs, 0.8% monkey |
| De Aluja *et al.*, (1996) [13] | pigs | 16 | *T. solium* | Not Reported |
| Fan *et al.*, (1996) [43] | pigs | 19 | *T. s asiatica* | 1–2% |
| Kaur *et al.*, (1995) [44] | pigs | 12 | *T. solium* | 0–600 cysts per pig |
| Fan *et al.*, (1994) [45] | pigs, dogs, cats, goats, cattle | 3 pigs, 4 dogs, 4 cats, 2 goats, 1 calf | *T. solium* | 0.8% pig, 0.3% dog |
| Eom *et al.*, (1992) [46] | pigs, cattle | 16 pigs, 2 calves | *T. s. asiatica* | 0–0.7% pigs, calf rate Not Reported |
| Fan *et al.*, (1992a) [47] | pigs, cattle | 6 pigs, 1 calf | *T. s. asiatica* | 11% pigs, 6% calf |
| Fan *et al.*, (1990) [48] | pigs | 38 | *T. s. asiatica* | 0.27–27.2% |
| Fan *et al.*, (1989) [49] | pigs, cattle, goats | 8 pigs, 1 calf, 2 goats | *T. s. asiatica* | 0.6–5.6% pigs, 0.03% calf, 0.02% goat |

## 3.4 Synthesis of results

Several animal experimental models have been used to study cysticercosis with varying degrees of success [16, 21, 49] using varying sample sizes and ages of experimental animals. We found that researchers have used pigs, cattle, gerbils, voles, mice, rats, hamsters, monkeys, cats, dogs, sheep, and goats as experimental animals using different *Taenia* spp., with some experiments using a combination of these animals in the experimental models. The *Taenia* spp. used in

**Table 2. Description of the animal models used in the included studies for *T. saginata*.**

| Author | Animals used | Sample size | Cysticerci recovery rate/density |
|---|---|---|---|
| Dorny *et al.*, (2017) [50] | pigs, cattle | 5 pigs, 1 calf | 0% pigs, calf rate Not Reported |
| Kandil *et al.*, (2013) [51] | mice | 25 | 1–5 cysts per mouse |
| Lopes *et al.*, (2011) [52] | cattle | 25 | 0.01–12.55% |
| Scandrett *et al.*, (2009) [53] | cattle | 42 | 0–5.42 cysts per 100gram of tissue |
| Minozzo *et al.*, (2002) [1] | cattle | 5 | 0.01–1.43% |
| Oryan *et al.*, (1998) [54] | cattle | 11 | 0.6–14 cysts per 10gram of tissue |
| Bogh *et al.*, (1996) [55] | cattle | 24 | 0–37 cysts per animal |
| Fan *et al.*, (1992b) [56] | pigs, cattle | 7 pigs, 1 calf | 36% pigs, 3% calf |
| Smith *et al.*, (1991) [57] | cattle | 15 | 0–52 cysts per animal |
| Geerts *et al.*,(1981) [58] | sheep, cattle | 8 sheep, 2 calves | Not Reported |

these experiments included *T. solium*, *T. saginata*. *T. s. asiatica*, *T. asiatica* and *T. crassiceps*. One study involving mice [41] and one involving rats [21] used a combination of two *Taenia* spp. in which the cysticerci recovery rate was not reported. Our findings revealed that the cysticerci recovery rate in all the studies was low. Thirty-six studies (75%) did not report the sex of the animals used. Two studies used male animals only (4.2%) while four studies used female animals only (8.3%). Six studies used a combination of male and female animals (12.5%). In most of these studies, breed or strain susceptibility to *Taenia* experimental infection was not assessed, even in studies that reported variations in cysticerci recovery rate due to breed, the results were not analysed statistically to determine whether statistical significance existed due to breed or strain of the animals used.

**3.4.1 Animal models used to study cysticercosis.** *3.4.1.1 Taenia solium.* A higher infection rate was seen in the immunosuppressed institute of cancer research (ICR) female mice (80%) as opposed to immuno-suppressed ICR males with an infection rate of 50%. Furthermore, male mice had a lower cysticerci recovery rate (0.05%) while females showed a 0.26% cysticerci recovery rate when infected with *T. solium* [2]. Higher infection rates were observed for *T. solium* (57–75%) in immunosuppressed male ICR mice following intravenous injection of oncospheres. In some studies, infection rate, cyst burden, and antibody-antigen levels were shown to increase with an increase in the dose of oncospheres inoculated in the animal [16, 23]. However, no correlation was found between the antibody concentration and the number of cysticerci recovered [33].

Additionally, lower doses of oncospheres used to achieve infection lead to a higher infection efficiency of 5.6% as opposed to 3.6% when a higher oncosphere dose of *T. solium* was surgically implanted in the subarachnoid space of piglets [23]. However, infection dose did not affect the development of cysticerci in the brain of rats after inoculation with *T. solium* activated oncospheres [24], though an increase in antigen titres due to an increase in the number of cysts was detected [28]. In contrast, the recovery rate of cysticerci decreased with an increase in the dose of *T. solium* eggs used [35]. Moreover, [37] demonstrated that pigs that ingested a lower dose of *T. solium* eggs had a 10% development of metacestodes as opposed to 0.75% in

**Table 3. Description of the animal models used in the included studies for *T. asiatica* and *T. crassiceps*.**

| Author | Animals used | Sample size | Taenia spp. Used | Cysticerci recovery rate/density |
|---|---|---|---|---|
| Chung et al., (2006) [59] | gerbils | 14 | *T. asiatica* | 0.1–3.2% |
| Ito *et al.*, (1997b) [60] | mice | 29 | *T. asiatica* | 5–202 cysts per mouse |
| Miyaji *et al.*, (1990) [5] | mice, gerbils, voles, rats, dogs | 22 mice, 26 gerbils, 24 voles, 4 rats, 8 dogs | *T. crassiceps* | 5–86 cysts per animal |

**Table 4. Description of the *Taenia* eggs as infecting material and the dose using the oral route in the studies for *T. solium*.**

| Author | Dose |
|---|---|
| Gomez-Puerta *et al.*, (2018) [22] | 52–312 |
| Borkataki *et al.*, (2013) [25] | 100000 |
| da Silva *et al.*, (2012) [26] | 200000 |
| Maravilla *et al.*, (2008) [29] | 50000 |
| Garrido *et al.*, (2007) [30] | 100000 |
| Soares *et al.*, (2006) [33] | 200000 |
| Nguekam *et al.*, 2003 [35] | 1000–100000 |
| Santamaria *et al.*, (2002) [37] | 10–100000 |
| De Aluja *et al.*, (1999) [40] | 100000 |
| De Aluja *et al.*, (1996) [13] | 100000 |
| Kaur *et al.*, (1995) [44] | 5000–20000 |
| Fan *et al.*, (1994) [45] | 2000–10000 |

**Table 5. Description of the infecting material, their dose and infection route, which includes intravenous (IV); subcutaneous (SC); intramuscular (IM); intraperitoneal (IP); intraarterial (IA); intracranial (IC); intraduodenal (ID);oral (OR) and surgical (S), used in the studies for *T. solium*.**

| Author | Infecting material | Route | Dose |
|---|---|---|---|
| Carmen-Orozco *et al.*, (2021) [20] | oncospheres | IC | 120 |
| Palma *et al.*, (2019) [21] | oncospheres, postoncospheres | IC | 10–180 |
| Mejia *et al.*, (2019) [17] | oncospheres | IC, OR | 500 IC, 20000 OR |
| Alroy *et al.*, (2018) [16] | oncospheres | IA | 10000–50000 |
| Fleury *et al.*, (2015) [23] | oncospheres | S | 500–1000 |
| Verastegui *et al.*, (2015) [24] | oncospheres | IC | 10–40 |
| Deckers *et al.*, (2008) [28] | whole proglottids | OR | Not Reported |
| Liu *et al.*, (2002) [36] | oncospheres | IM, IV | 500 |
| Verastegui *et al.*, (2002) [38] | oncospheres | IM | 250–2500 |
| Verastegui *et al.*, (2000) [14] | oncospheres | IM, IP, IV, ID | 250–2500 |
| Wang *et al.*, (1999) [2] | oncospheres | SC, IV | 500–5000 |

**Table 6. Description of the infecting material, dose and route such as oral (O) and intraperitoneal (IP), used in the studies for *T. saginata*.**

| Author | Infecting material | Route | Dose |
|---|---|---|---|
| Dorny *et al.*, (2017) [50] | eggs | OR | 5000–30000 |
| Kandil *et al.*, (2013) [51] | oncospheres | IP | 5000 |
| Lopes *et al.*, (2011) [52] | eggs | OR | 20000 |
| Scandrett *et al.*, (2009) [53] | eggs | OR | 10–10000 |
| Minozzo *et al.*, (2002) [1] | eggs | OR | 20000 |
| Oryan *et al.*, (1998) [54] | eggs | OR | 5000–50000 |
| Bogh *et al.*, (1996) [55] | eggs | OR | 30000 |
| Fan *et al.*, (1992b) [56] | eggs | OR | 1000–10000 |
| Smith *et al.*, (1991) [57] | eggs | OR | 10–10000 |
| Geerts *et al.*, (1981) [58] | eggs | OR | 500–10000 |

**Table 7. Description of the infecting material, dose and route such as subcutaneous (SC); oral (OR); intraperitoneal (IP) and intravenous (IV) used in the studies for *T. s asiatica*.**

| Author | Infecting material | Route | Dose |
|---|---|---|---|
| Peng *et al.*, (2009) [27] | oncospheres | SC | 5000 |
| Linghu *et al.*, (2007) [31] | eggs | OR | 120000 |
| Fan *et al.*, (2006) [32] | eggs | OR | 1000–30000 |
| Chang *et al.*, (2005) [34] | oncospheres | SC, IP | 20000–40000 SC |
| | | | 18600 IP |
| Wang *et al.*, (1999) [2] | oncospheres, eggs | SC, IV | 500–5000 |
| Ito *et al.*, (1997a) [41] | oncospheres, eggs | OR | NR |
| Chung and Fan (1996) [42] | eggs | OR | 1500–30000 |
| Fan *et al.*, (1996) [43] | oncospheres | IV | 5000–10000 |
| Eom *et al.*, (1992) [46] | eggs | OR | 25000–890000 |
| Fan *et al.*, (1992a) [47] | eggs | OR | 10000 |
| Fan *et al.*, (1990) [48] | eggs | OR | 1000–100000 |
| Fan *et al.*, (1989) [49] | eggs | OR | 1000–380000 |

**Table 8. Breed, *Taenia* spp. and age of animals used in experimental models.**

| Animals used | Breed | | Age (days) | Taenia spp. |
|---|---|---|---|---|
| Pig | i. Duroc-Yorkshire- Landrace | vii. Yorkshire- Landrace | | *T. solium* |
| | | | | *T. s asiatica* |
| | ii. Yorkshire | viii. Landrace | 5 to 730 | *T. saginata* |
| | iii. Hampshire | ix. Landrace-Duroc-Hampshire | | |
| | iv. Duroc | x. Landrace- Duroc | | |
| | v. Landrace- Small Ear Miniature | xi. Landrace-Hampshire | | |
| vi. Small Ear Miniature | | | | |
| Cattle | Holstein | Hereford | | *T. solium* |
| | Crossbreed | Angus | 4 to 570 | *T. s asiatica* |
| | Jersey | | | *T. saginata* |
| Gerbils | i. *Meriones unguiculatus* | | 35 to 98 | *T. asiatica* |
| | | | | *T s. asiatica* |
| Mice | i. Balb/CAnN | iv. ICR | | *T. solium* |
| | ii. C57BL/6N | v. SCID | | *T. s asiatica* |
| | iii. C3H/HeN | vi. CB17-scid | 35 to 84 | *T. saginata* |
| | | | | *T. crassiceps* |
| Voles | i. *Clethrionomys rufocanus bedfordiae* | | Not Reported | *T. crassiceps* |
| Rats | i. Holtzman | | | *T. solium* |
| | ii. Wistar | | 6 to 35 | *T. saginata* |
| | | | | *T. crassiceps* |
| Hamsters | i. Golden | | 21 | *T. s asiatica* |
| Monkeys | i. *Macaca cyclopis* | | 360 | *T. s asiatica* |
| Cats | NS | | Not Reported | *T. solium* |
| Dogs | i. Mongrel | | 7 to 150 | *T. solium* |
| Sheep | i. Soay | | Not Reported | *T. saginata* |
| | ii. Texel | | | |
| | iii. Four-horned | | | |
| Goats | i. Saanen | | 5 to 13 | *T. solium* |
| | | | | *T. s asiatica* |

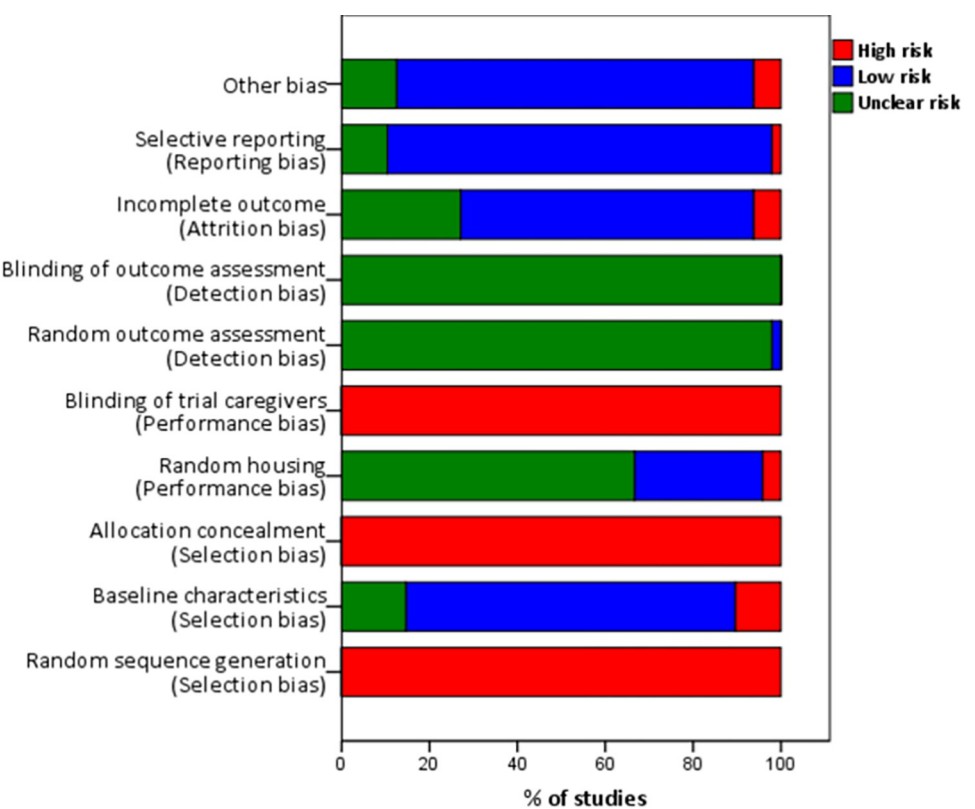

**Fig 2. Risk of bias in included studies using the SYRCLEs risk of bias tool.**

pigs that received a higher dose, however, the higher the dose, the more the larvae remained vesicular and infective for a longer duration.

The Non-immunosuppressed ICR, Balb/c, and C3H mice were not susceptible to oncospheres of *T. solium*. Nevertheless, following immunosuppression, the Balb/c, C3H, and C57 mice were susceptible to the oncospheres of *T. solium* with infection rates of 50%, 60%, and 100%, respectively, and the cysticerci recovery rates of 0%, 0.43%, and 0.12% respectively [2]. Besides, the normal C57BL/6N mice were also found to have a high infection rate of 80% with a cysticercus recovery rate of 0.02 to 2.4% [2].

Studies for *T. solium* using single breed of pigs namely Landrace, Landrace crossed with Yorkshire (LY) and Landrace crossed with Duroc and Hampshire (LDH) showed cysticerci recovery rates of 0.2 to 81.93%, 0 to 5.4%, and 0 to 0.71% respectively (Table 1). Infection with *T. solium* in 10-day old rats showed an infection rate of 83%, while 18-day old rats had an infection rate of 66%, whereas the 26-day old rats had the lowest infection rate of 25%. This study showed that the number of cysticerci detected reduced with rat age [24]. In yet another study, older pigs demonstrated some degree of resistance to infection [13].

On the other hand, [28] demonstrated that the number of viable cysts reduced with an increase in pig age (i.e. 5 months old pigs compared with 1 and 3 months old pigs). In this study, detected antigens were high in pigs infected at 1 month, thus demonstrating higher susceptibility to infection in younger pigs.

*3.4.1.2 Taenia saginata.* [1], found older cattle to be more resistant to *T. saginata* egg infection showing a lesser number of cysticerci and a higher number of calcified cysts. The cysticerci recovery rate was not reported in one of the studies [58]. As expected, an increase in the

number of eggs used to infect the experimental animals increased the cysticerci recovery rate [57, 58]. [51] found an increase in serum globulins and a marked decrease in the albumin globulin ratio in the mice infected with *T. saginata* oncospheres. They further suggested that female BALB/c mice can be used as experimental animals for studying the host immune response in vaccine development trials. [50] exposed pigs to *T. saginata* eggs experimentally and no cysticerci were recovered from the experimental pigs, this resulted in negative serological tests for *T. solium*. In addition, [55], demonstrated a higher total number of cysts in calves infected with a single dose of *T. saginata* eggs compared to calves that were trickle infected by weekly oral administration of eggs for 12 weeks. Interestingly, higher cysticerci numbers were recovered from cattle infected with *T. saginata* eggs in regions that are not officially examined as they are considered non-preferential sites for cysticercus bovis [52]. Similarly, [53] found cysticerci in the non-traditional sites following oral inoculation of cattle with *T. saginata* eggs. [54] Oryan et al. (1998), found that the age of the animals and dose of *T. saginata* eggs influenced clinical signs and pathological changes in the calves infected with *T. saginata* eggs.

*3.4.1.3 Taenia saginata asiatica.* Susceptibility and cyst recovery of *T. s. asiatica* oncospheres in immunosuppressed male ICR mice following venous injection was assessed and lower infection rates were observed (14–20%). In the study for *T. s. asiatica*, C3H/HeN mice had the highest cysticerci recovery rate compared to the BALB/CAnN and C57BL/6N mice [27]. In the study conducted by [49] Fan *et al.* (1989) for *T. s. asiatica*, the Landrace crossed with the small ear miniature pigs (L-SEM) had the highest cysticerci recovery rate of 5.6% as opposed to 1.7% for the small ear miniature (SEM) and 0.03% for the Duroc-Yorkshire-Landrace cross pigs (DYL). In Contrast to the later study by [48] a cysticerci recovery rate of 27.1% was found for the SEM, 1.7% L-SEM, and 0.27% DYL for *T. s. asiatica*. However in a study for *T. s. asiatica*, a high infection rate was seen in the SEM (80%) with a cysticerci recovery rate of 36%, and no infection was seen in the L-SEM [56] (Fan *et al.*, 1992b). Moreover, the SEM pigs were found as a favourable host for *T. s. asiatica* with a cysticerci recovery rate of 0.005 to 22% and an infection rate of 75 to 100% as opposed to the L-SEM pigs that had a cysticerci recovery rate of 1.1 to 5.6% and an infection rate of 83 to 100%. In addition, the DYL pigs had an infection rate of 100% and cysticerci recovery rate of 0.06 to 0.3%. [42], infected pigs with *T. s. asiatica* and found the cysticerci recovery rate of 1.4 to 22.9% in the SEM whereas the L-SEM had a cysticerci recovery rate of 16.1%.

*3.4.1.4 Taenia asiatica.* In addition, female severe combined immunodeficiency (SCID) mice developed cysticerci in the peritoneal cavity or under the skin after infection with oncospheres of *T. asiatica* whereas males did not [60]. In the study conducted by [59], cysticerci were recovered from SC inoculation of gerbils with oncospheres. However, no cysticerci were recovered from gerbils orally inoculated with eggs. In this study, the infectivity of the cysticerci was evaluated were a total of seven adult worms were recovered from the two human volunteers who ingested five cysticerci after 120 days post infection.

*3.4.1.5 Taenia crassiceps.* Two out of twelve mice became infected after oral inoculation with 100 eggs whereas no cysticerci developed in mice following inoculation with 500 or 5000 eggs of *T. crassiceps* [5]. In this study breed difference was not reported, voles had an infection rate of 50%, gerbils 34.6%, mice 17% after inoculation with *T. crassiceps* eggs [5].

*3.4.1.6 Models inducing neurocysticercosis.* Rats developed NCC following IC and OR inoculation with activated oncospheres and postoncospheres of *T. solium* [17, 21]. Interestingly, the dose of *T. solium* activated oncospheres affected the infection efficiency, where an increase in infection efficiency of 5.4% was seen in pigs that received a lower dose as opposed to an infection efficiency of 3.6% in pigs that received a higher oncosphere dose after surgical implantation of oncospheres in the cerebral subarachnoid space of the piglets [23]. Nonetheless, vesicular cysts were found in the brains of pigs following oral infection with *T. solium* eggs

[13, 40]. In addition, cyst burden was high in the brains of pigs inoculated with a high dose of activated oncospheres in the common carotid artery as opposed to the pigs inoculated with a lower dose of *T. solium* [16]. In another study, rats were inoculated intracranially (extraparenchymally and intraparenchymal) with *T. solium* activated oncospheres to induce NCC. In this study, the route of infection and infection dose did not affect the proportion of rats that developed cysticerci in the brain [24]. Following IC inoculation of rats with *T. solium* oncospheres, an increased expression of genes associated with proinflammatory response and fibrosis related proteins was observed in the brain tissue of infected rats four months after infection [20].

**3.4.2 Infecting material and route of infection used in the animals' experimental models.** Various routes of infection were used in the models with some studies using a combination of the various routes and infecting material (Fig 3). The routes used to induce infection in these studies included the oral, intravenous, subcutaneous, intramuscular, intraperitoneal, intraarterial, intracranial, surgical, and intraduodenal routes. The infecting material used included eggs, proglottids, and oncospheres. Briefly, the *T. solium*, *T. saginata*, *T. s. asiatica* and *T. asiatica* eggs were obtained from gravid proglottids collected from individuals habouring the adult worms after treatment followed by a purge. Gravid segments were repeatedly washed, centrifuged, or triturated in a pestle and mortar to obtain the eggs. The collected eggs were exposed for 10 minutes to 0.75% sodium hypochlorite at 4 degrees celsius for oncosphere hatching in the studies. Additionally, gerbils and mice were SC, IP, and OR inoculated with oncospheres and later euthanased to obtain the cysticerci of *T. s. asiatica* [34].

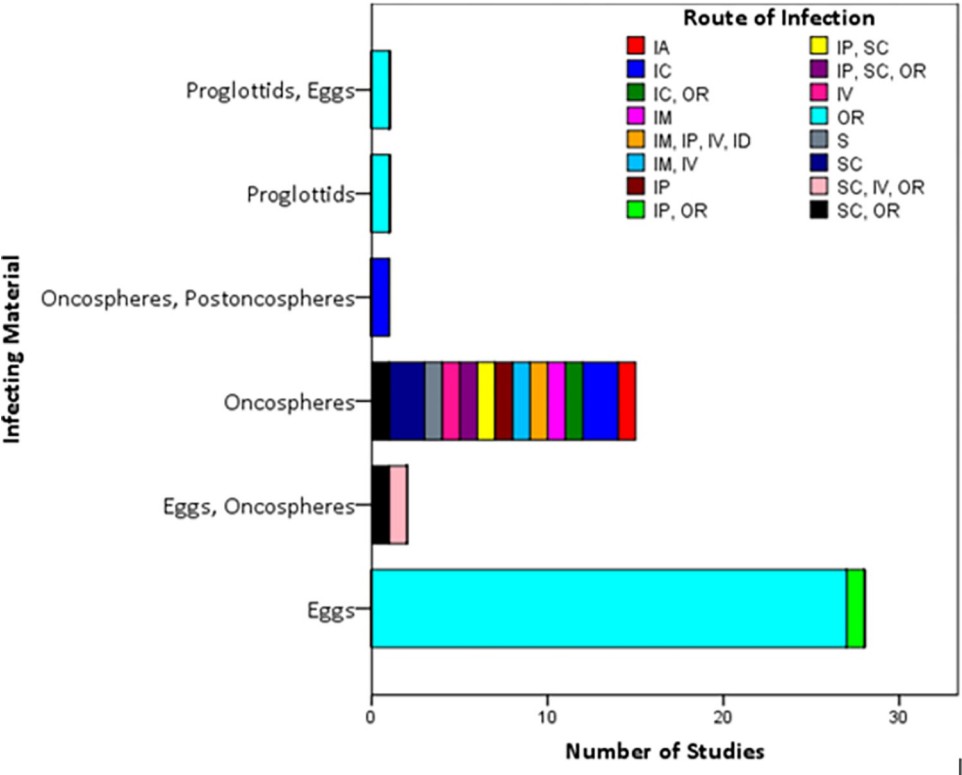

**Fig 3. Chart showing route (s) of infection versus material (s) used to infect experimental animals in individual studies.** OR = oral; IV = intravenous; SC = subcutaneous; IM = intramuscular; IP = intraperitoneal; IA = intraarterial; IC = intracranial; ID = intraduodenal; S = surgical.

Our review has shown that that the oral route was the most frequent route used in 29 out of 48 studies (60.4%) using eggs alone at various doses as the infecting material. The use of proglottids alone, both proglottids and eggs, both oncospheres and postoncospheres were reported in one study in each case (i.e. each representing 2.1% of the total included studies). The use of both eggs and oncospheres was reported in two studies, representing 4.2% of the included studies. Oncospheres were used as infecting material in 15 studies representing 31.3% of the included studies. Nine studies used a combination of routes to achieve infection in the various animals studied (Fig 3).

No cysticerci were detected via oral infection method using eggs or oncospheres but a high infection rate was achieved in mice via subcutaneous route using oncospheres in both *T. s. asiatica* and *T. solium* [41]. Also, [60] suggested that the intraperitoneal inoculation of mice with oncospheres yields a higher cysticerci recovery rate than the subcutaneous inoculation. However, infection was achieved in gerbils subcutaneously injected with hatched oncospheres whereas no infection was established when oncospheres were orally fed to the gerbils [59]. Similarly, oncospheres of *T. s. asiatica* were inoculated subcutaneously in SCID mice and yielded a higher cysticerci recovery rate of 0.1 to 1.1% while intraperitoneal inoculation yielded a lower cysticerci recovery rate of 0.3% [34]. Furthermore, some mice inoculated with oncospheres of *T. solium* via the intravenous route developed cysticerci, whereas no cysticerci developed in mice inoculated with oncospheres through the intramuscular route [36]. Gerbils and voles were observed to have higher infection rates than ICR mice after oral inoculation with eggs of *T. crassiceps* [5]. In contrast, the route of infection did not affect the development of cysticerci in the rats after inoculation with *T. solium* oncospheres [24]. The dose of infecting material used in the various studies are shown in Tables 4–7 while breed, *Taenia* spp. and age of animals used in the experimental models is shown in Table 8.

*3.4.2.1 Description of the infecting material, dose and route used in the studies for T. asiatica and T. crassiceps.* In the study conducted by [59], *T. asiatica* eggs and oncospheres were inoculated in experimental gerbils using the subcutaneous and oral route. However, in this study, the doses of eggs and oncospheres were not reported. Additionally, [60] infected the experimental mice with 50,000 oncospheres of *T. asiatica* using the intraperitoneal and subcutaneous route. [5] infected the experimental mice, gerbils and voles using 100–5000 eggs of *T. crassiceps* using the oral route.

**3.4.3 The strengths and weaknesses of the animal models.** *3.4.3.1 Strengths.* Pig models have shown that experimentally infected pigs serve as the good model to study cysticercosis as the disease in pigs mimics the human form of the disease. Pig models using whole proglottids have an advantage of mimicking the natural method in which pigs get infected. Moreover the infection date and dosage is known as opposed to natural infection. The use of monkeys to study cysticercosis has an advantage as the disease in monkeys may be useful to study the human disease due to immense similarities of monkeys with humans. Rodents such as mice and rats were used in severel models, these animals have an advantage of easier of handling and the entire sequence of their genome is known which may aid in making sound interpretations of experimental results.

Some of these experiments have demonstrated that the cysticerci established can be used to infect natural and other alternative definitive hosts for the establishment of adult worms [5, 34, 59] and can thus be used for future experiments especially in *T. solium* experimental studies where the collection of the tapeworm from taeniasis positive humans is quite challenging. Additionally, these studies have shown some differences in the establishment of cysticerci and resistance to infection due to sex, age, breed, and route of inoculation of the infecting material.

*3.4.3.2 Weaknesses.* Large animals such as cattle, pigs and goats are difficult to handle in intensive care units because they require trained personel to handle and thus make the entire

experimental process laborious which may lead to several experimental errors. Laboratory mice and rats are not susceptible to natural *T. solium* infection making their models not good to study human cysticercosis and NCC. Moreover, some models used immunosuppressed animals to achieve experimental infection, this may present some limitations to understanding the host-parasite interaction under the normal physiological status of the host. Additionally, several experimental animals were inoculated using different routes which are not their natural routes of infection, although infection was achieved in these models, results may not be extrapolated to understanding the normal pattern of how the hosts gets infected.

In most (if not all) studies there was insufficient reporting and non-usage of methods to reduce bias, such as sample size calculation. None of the studies reported how the sample size was calculated. Therefore, the power of the experimental studies (i.e. probability to detect treatment effect if it existed) could not be established, which may compromise the process of detecting the difference between experimental groups. Nonetheless, studies with low power were included in this systematic review because we wanted to give a full overview of animal models that have been used to study cysticercosis. In addition, it was not indicated clearly in the studies whether animals were selected at random for outcome assessment and whether outcome assessors were blinded from knowing the intervention that each animal received and thereby introducing detection bias. Furthermore, improvements are needed throughout experiments from random housing allocation and sequence generation. Moreover, the trial caregivers knew the intervention that each animal received and could thus introduce performance bias. Even though breed, sex, and age differences were reported in some of the studies, these results were not analysed statistically to determine if any significant difference existed among these risk factors of cysticercosis and thus the conclusions made may be questionable.

## 4.0 Discussion

In total, 12 animal species were used in the experimental models involving 5 *Taenia* spp. In these studies, infection was established in the animals through inoculation with eggs, oncospheres, and proglottids using various routes. However, the cysticerci recovery rate was low in all the cases. Our findings revealed that the dose of eggs and oncospheres, used affected the rate of infection with few contrasting results. The variations in the establishment of cysticerci in the various experiments could be due to the genetic differences of the parasites used to infect the animals as the parasites were obtained from humans and animals with different genetic makeups [29].

In a few studies that reported sex differences in the cysticerci rates, females animals were shown to be more susceptible to infection as opposed to male animals. This suggests an influence of the sex hormones in susceptibility and resistance to infection by *Taenia* spp. [63, 64]. Although not statistically analysed, our findings revealed that breed, sex, and age of the animal had an influence on the cysticerci recovery rate in the infected animals. The lower infection and cysticerci recovery rates in older animals may suggest that parasites get destroyed as the animal grows by the host's innate and adaptive immune system [13].

The route of infection is an important parameter that should be considered when establishing infection in experimental animals using eggs, oncospheres, or proglottids. The results show that higher infection rates were obtained after subcutaneous inoculation of oncospheres as opposed to any other route of infection. Moreover, the cysticerci recovery rate was high in animals after subcutaneous inoculation with oncospheres. On the other hand, a higher cysticerci recovery rate was obtained when the intraperitoneal route was used as opposed to the subcutaneous route after oncosphere inoculation. To induce NCC, oncospheres were inoculated through the IC and surgical methods, and thus any other method of inoculation may not

yield the desired results. Besides, eggs were orally given to experimental animals and NCC was established in very few studies whereas oncospheres were inoculated IA and cysticerci were recovered in the brains of infected animals. However, if any other method other than oral, IA, IC, and surgical is to be used, it should be investigated whether it can induce NCC. Most investigators could not induce NCC in the large experimental animals because of the uncertainty and variability of the oral infection efficacy. Moreover, the cost of purchasing large animals and long-time maintenance of the animals as opposed to laboratory animals is another limiting factor [24]. Although rats and mice are not the natural hosts of *T. solium* that cause NCC in humans, they can be used as models to study the human disease following IC inoculation in which the recovered cysticerci have characteristics similar to the ones observed in humans and pigs [24]. The effect of the route of infection has been shown in other studies like that of [65] where the course of infection of *Brucella melitensis* was investigated following inoculation of C57BL/6 mice using three different routes of inducing disease.

The other routes of infections are less convenient and many authors preferred using the oral route which has its challenges. Eggs were orally given to the experimental pigs in most of the studies and the results varied based on the dose and age of the animals. Several studies reported a low infection rate following oral inoculation of experimental animals. These low infection rates could be attributed to the fact that eggs were kept for a long time before being used to infect experimental animals [26]. Moreover, eggs were removed from the faecal material or proglottids which may play a role in achieving infection. Further, most studies did not assess egg viability before infecting the experimental animals. Therefore, they may have infected animals with eggs of questionable viability leading to low infection rates. In addition, the techniques in many studies are not standardized leading to significant variability in the models used [14]. Another limiting factor in infecting natural hosts could be the presence of maternal antibodies and thus future studies should aim at testing all experimental animals to ensure that they are free from these antibodies if high infection rates are to be achieved [35]. In some studies, eggs of the *Taenia* parasites were exposed to animals in which natural infection does not occur and could be one of the reasons for the lower infection rates obtained [50].

In our opinion, the number of animals used in several studies was not adequate. Some pig, cattle, rat, monkey, cat, dog, and goat studies used 1 to 4 animals which were not sufficient enough to detect any differences due to the induction of infection. Besides, it is not feasible to divide animals into control and treatment groups with inadequate sample sizes. However, mice and rat studies had a fairly good sample size as opposed to most of the large animal studies. Limited sample sizes used in some models coupled with the fact that animals were kept in well-controlled experimental conditions which are different from field conditions especially in large animal models could be another factor causing low infection rates. In natural conditions, factors such as poor nutrition, as opposed to well-fed experimental animals, may impair the immune systems of the animals and thus make them more susceptible to infection [50]. To be able to determine whether or not a significant difference exists between means or proportions observed in comparison groups, appropriate sample size is required because of its effect on statistical power. We propose that future experimental models focus on avoiding the use of inadequate sample size and other design issues. This will help investigators to make sound conclusions as it is wasteful and inappropriate to conduct a study with inadequate power [66]

## 5.0 Conclusion

Overall, this systematic review shows that several animal models have been used to study cysticercosis in animals using *Taenia* spp. with varying degrees of success. The cysticerci recovery rate differences were attributed to breed, age, sex, and the routes of inoculation used to

establish infection. The poor reporting of some methodological details in the animal experiments as revealed by this review may lead to a lack of repeatability of the models and may hinder drawing well-founded conclusions from some of the studies conducted. Therefore, future animal models should be of high methodological quality to eliminate bias if our current knowledge of cysticercosis is to be improved.

## Supporting information

**S1 Checklist. PRISMA-P 2015 checklist.**
(DOCX)

## Acknowledgments

I wish to thank Elizabeth Thandiwe Chimera Sitali and Madalisto Chelenga for helping me search for the articles regardless of their busy schedule.

## Author Contributions

**Conceptualization:** Muloongo C. Sitali, Chummy S. Sikasunge, Kabemba E. Mwape, Isaac K. Phiri.

**Data curation:** Muloongo C. Sitali.

**Formal analysis:** Muloongo C. Sitali.

**Investigation:** Muloongo C. Sitali.

**Methodology:** Muloongo C. Sitali, Racheal Mwenda.

**Supervision:** Veronika Schmidt, Chummy S. Sikasunge, Kabemba E. Mwape, Martin C. Simuunza, Clarissa P. da Costa, Andrea S. Winkler, Isaac K. Phiri.

**Validation:** Muloongo C. Sitali.

**Writing – original draft:** Muloongo C. Sitali.

**Writing – review & editing:** Muloongo C. Sitali, Veronika Schmidt, Racheal Mwenda, Chummy S. Sikasunge, Kabemba E. Mwape, Martin C. Simuunza, Clarissa P. da Costa, Andrea S. Winkler, Isaac K. Phiri.

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
