## [Decision Letter · Decision Letter 0]

28 Apr 2022

PONE-D-22-01651Experimental animal models and their use in understanding cysticercosis: A systematic reviewPLOS ONE

Dear Dr. Sitali,

Thank you for submitting your manuscript to PLOS ONE. After careful consideration, we feel that it has merit but does not fully meet PLOS ONE’s publication criteria as it currently stands. Therefore, we invite you to submit a revised version of the manuscript that addresses the points raised during the review process.

We look forward to receiving your revised manuscript.

Kind regards,

Adler R. Dillman, Ph.D.

Academic Editor

PLOS ONE

Journal Requirements:

Reviewers' comments:

Reviewer's Responses to Questions

**Comments to the Author**

1. Is the manuscript technically sound, and do the data support the conclusions?

Reviewer #1: Partly

Reviewer #2: Partly

2. Has the statistical analysis been performed appropriately and rigorously? 

Reviewer #1: No

Reviewer #2: N/A

3. Have the authors made all data underlying the findings in their manuscript fully available?

Reviewer #1: Yes

Reviewer #2: Yes

4. Is the manuscript presented in an intelligible fashion and written in standard English?

Reviewer #1: No

Reviewer #2: Yes

5. Review Comments to the Author

Reviewer #1: The article entitled “Experimental animal models and their use in understanding cysticercosis: A systematic review” reported the results of a systemic review without meta-analysis on the question: What are the experimental animal models and their use in understanding cysticercosis. It was a good start that the authors mentioned NCC, which would be interesting for the relevant field if the question is to identify animal models for NCC. Unfortunately, the authors didn’t form an exclusive question, thus failing to write a scientifically sound review. The purpose of a systematic review is to draw conclusions based on the evidence to answer the well-defined and narrow question. On another note, it appears to me that the authors are often confused about different species of Taenia and the diseases that may cause in humans. Also the paper was not carefully written including the references provided. Thus, I suggest the authors reconstruct their questions to improve the review.

Here are a few examples:

Line65: “T. asiatica, T. solium and T. saginata share the same intermediate host.” This is scientifically wrong. But later in Line69, authors stated “……share the same definitive host”

Line169, “infecting material used (eggs, …. cysticerci)” cysticerci should not be used for infection of the purpose.

Line244- 248, is under the subtitle Taenia solium, but, as a matter of fact, reference#57 and 58 are about T. saginata. Thus, the authors finding is not valid.

References: #8, the year is 2020, should remove 2007; #57, missing journal name

Reviewer #2: It is an interesting article where a systematic bibliographical review is carried out, of the different animal models that have been developed to study the interaction between Taenia spp. and its intermediate host. It should be accepted with some major changes

My biggest criticism is in the section on the strengths and weaknesses of the animal models, should be highlighted as a tool to understand the pathogenesis of neurocysticercosis. As the reproducibility of the animal model, the clinical picture and pathology that develops. As for example, the use of pigs to study NCC has the advantage of being the natural host, however it is difficult to have a good reproducibility of the experimental infection of the central nervous system, while the use of rats presents a high reproducibility, but has the disadvantage of not being the natural host. The use of immunosuppressed animals has many limitations to understand the host-parasite interaction. To have animal model with characteristic of clinical symptoms and pathology similar to human , it is help to study the disease.

My minor criticism is in Table 1, the study by Alan Mejia Maza (2018) should be included, where they use the NCC rat model, in which they compare two routes of infection (intracranial and oral).

On the other hand, in Table 1, in the published article by Verastegui et al. (2015), the number of cysts obtained in rats with NCC is reported.

6. PLOS authors have the option to publish the peer review history of their article (what does this mean?). If published, this will include your full peer review and any attached files.

Reviewer #1: No

Reviewer #2: No

---

## [Author Response · Author response to Decision Letter 0]

15 Jun 2022

Review Comments to the Author

Reviewer #1: The article entitled “Experimental animal models and their use in understanding cysticercosis: A systematic review” reported the results of a systemic review without meta-analysis on the question: What are the experimental animal models and their use in understanding cysticercosis. It was a good start that the authors mentioned NCC, which would be interesting for the relevant field if the question is to identify animal models for NCC. Unfortunately, the authors didn’t form an exclusive question, thus failing to write a scientifically sound review. The purpose of a systematic review is to draw conclusions based on the evidence to answer the well-defined and narrow question. On another note, it appears to me that the authors are often confused about different species of Taenia and the diseases that may cause in humans. Also the paper was not carefully written including the references provided. Thus, I suggest the authors reconstruct their questions to improve the review.

Authors’ response: We note here the reviewer’s comments. As outlined by the reviewer, the main question we asked in this systematic review was: What are the experimental animal models and their use in understanding cysticercosis? Thus, we aimed at identifying animal models that have been used to understand cysticercosis (including neurocysticercosis), the infection caused by larvae of the tapeworm of the genus Taenia. Therefore, the systematic review investigated experimental animal models and their use in understanding both cysticercosis and NCC. We understand that NCC is what has made Taenia solium be of public health concern and that it would have been more specific had we only restricted the review to NCC. However, this systematic review was guided by the following research questions: 

1) What animal experimental models (species of animals used and method of inducing disease) have been used to understand cysticercosis? 

2) What are the strengths and weaknesses of these experimental models?

3) What sample sizes have been used?

4) Have the models contributed significantly to the understanding of cysticercosis?

A meta-analysis was not done because characteristics of study populations (age, sex, breed or strain of animals) and interventions used in the selected animal models were too dissimilar to combine and therefore opted for a narrative synthesis of the findings.

The following where the data items

Name of species of animals used, number of species of animals used, Taenia species used, age of the animals used, number of the animals used, breed or strain of the animals used, sex, method of infection (oral, surgical or any other method employed in the study), infecting material used (eggs/proglottids/oncospheres) and the cysticerci recovery rate. 

Here are a few examples:

Line65: “T. asiatica, T. solium and T. saginata share the same intermediate host.” This is scientifically wrong.But later in Line69, authors stated “……share the same definitive host”

Authors’ response: The error of T. asiatica, T. solium and T. saginata sharing the same intermediate host has been corrected as observed by the reviewer (see Lines 68-70 of the revised manuscript). Revised sentence reads as:‘‘ The genus Taenia contains many species that infect humans and domestic animals. Of these, T. solium and T. asiatica eggs can infect pigs if they ingest eggs excreted from human tapeworm carriers whereas cattle serve as intermediate hosts for T. saginata ’’ 

Line169, “infecting material used (eggs, …. cysticerci)” cysticerci should not be used for infection of the purpose.

Authors’ response: We agree with the reviewer that the studies which used initial stage cysticerci to induce NCC be removed. The two studies were thus removed from the review as suggested and cysticerci deleted from all sections were it appeared (see table 3 in revised manuscript). Descriptive statistics was redone to reanalyse the proportions because of this deletion and the concerned references were deleted from the reference list. Therefore, the reference list was re-numbered.

Line244- 248, is under the subtitle Taenia solium, but, as a matter of fact, reference#57 and 58 are about T. saginata. Thus, the authors finding is not valid.

Authors’ response: The reference 57 and 58 were misplaced during the formatting process. The two references have been deleted from lines 244-248 and now been placed under the correct subtitle of T. saginata. (See Lines 275- 278 of the revised manuscript).

References: #8, the year is 2020, should remove 2007; #57, missing journal name

Authors’ response: Reference #8 year has been corrected to 2020 (see lines 571-573 of the revised manuscript); Reference #57 missing journal name has been included (Canadian Journal of veterinary research), see line 762-764 of the revised manuscript.

Reviewer #2: It is an interesting article where a systematic bibliographical review is carried out, of the different animal models that have been developed to study the interaction between Taenia spp. and its intermediate host. It should be accepted with some major changes

My biggest criticism is in the section on the strengths and weaknesses of the animal models, should be highlighted as a tool to understand the pathogenesis of neurocysticercosis. As the reproducibility of the animal model, the clinical picture and pathology that develops. As for example, the use of pigs to study NCC has the advantage of being the natural host, however it is difficult to have a good reproducibility of the experimental infection of the central nervous system, while the use of rats presents a high reproducibility, but has the disadvantage of not being the natural host. The use of immunosuppressed animals has many limitations to understand the host-parasite interaction. To have animal model with characteristic of clinical symptoms and pathology similar to human, it is help to study the disease.

Authors’ response: The section of weaknesses and strengths has been edited to incorporate the reviewers’ suggestions, (see lines 412-442 of the revised manuscript).

My minor criticism is in Table 1, the study by Alan Mejia Maza (2018) should be included, where they use the NCC rat model, in which they compare two routes of infection (intracranial and oral).

Authors’ response: The study by Alan M. Maza 2019 has been included in Table 1 and Table 5 and cited in the section for models inducing NCC (see line 323-326 in the revised version).The publication has also been added to the Reference list (see line number 607-609 in the revised version). Descriptive statistics was redone to reanalyse the proportions because of this addition and the reference list numbering edited (see edited reference list of the revised manuscript).

On the other hand, in Table 1, in the published article by Verastegui et al. (2015), the number of cysts obtained in rats with NCC is reported.

Authors’ response: The number of cysts obtained in the rats by Verastegui et al (2015), has been included in Table 1 (1-4 cysts per rat).

---

## [Editor Report · Decision Letter 1]

27 Jun 2022

Experimental animal models and their use in understanding cysticercosis: A systematic review

PONE-D-22-01651R1

Dear Dr. Sitali,

We’re pleased to inform you that your manuscript has been judged scientifically suitable for publication and will be formally accepted for publication once it meets all outstanding technical requirements.

Kind regards,

Adler R. Dillman, Ph.D.

Academic Editor

PLOS ONE

Additional Editor Comments (optional):

Thank you for your point-by-point response to reviewer concerns.
---

## [Editor Report · Acceptance letter]

8 Jul 2022

PONE-D-22-01651R1 

Experimental animal models and their use in understanding cysticercosis: A systematic review 

Dear Dr. Sitali:

I'm pleased to inform you that your manuscript has been deemed suitable for publication in PLOS ONE. Congratulations! Your manuscript is now with our production department. 

Kind regards, 

on behalf of

Dr. Adler R. Dillman 

Academic Editor

PLOS ONE